# Immunogenicity and Safety of Childhood Combination Vaccines: A Systematic Review and Meta-Analysis

**DOI:** 10.3390/vaccines10030472

**Published:** 2022-03-18

**Authors:** Bei Liu, Bing Cao, Chao Wang, Bingfeng Han, Tao Sun, Yudong Miao, Qingbin Lu, Fuqiang Cui

**Affiliations:** 1Department of Laboratorial Science and Technology, School of Public Health, Peking University, Beijing 100191, China; 1916387057@bjmu.edu.cn (B.L.); wchao@bjmu.edu.cn (C.W.); 2Key Laboratory of Cognition and Personality, Faculty of Psychology, Ministry of Education, Southwest University, Chongqing 400715, China; bingcao@swu.edu.cn; 3Department of Epidemiology and Biostatistics, School of Public Health, Peking University, Beijing 100191, China; hanbingfeng@pku.edu.cn; 4Department of Health Policy and Management, School of Public Health, Hangzhou Normal University, Hangzhou 311121, China; 20190086@hznu.edu.cn; 5Department of Health Policy and Management, College of Public Health, Zhengzhou University, Zhengzhou 450001, China; meldon@zzu.edu.cn

**Keywords:** combined vaccine, meta-analysis, infants, immunogenicity, safety

## Abstract

Background: Vaccination is considered the most effective and economical measure for controlling infectious diseases. Although combination vaccines are widely used worldwide, whether any of the combination vaccines is superior to each separate vaccine has yet to be established. This systematic review and meta-analysis aimed to summarize the available evidence on the effectiveness and safety of combination vaccines in children. Methods: A systematic search was conducted from database inception to August 20, 2021, in MEDLINE, Embase, Cochrane, and Scopus. Published randomized clinical trials (RCTs) and open-label trials of immunogenicity and safety of combined vaccines were selected. The results of the studies were quantitatively synthesized. Results: Overall, 25 articles met the inclusion criteria and were included in the meta-analysis. The results indicated that the combined diptheria–tetanus–acellular pertussis (DTaP)–hepatitis B virus (HBV)–*Haemophilus influenzae* type B (Hib) vaccine group had lower levels of anti-tetanus antibodies than the combined DTaP–HBV and separate Hib vaccinations group (SMD = −0.23; 95% CI: −0.42, −0.05; *p* = 0.013). Meanwhile, the combined DTaP–HBV–inactivated polio virus (IPV)–Hib vaccine group had higher levels of anti-pertussis (PT) and anti-filamentous hemagglutinin (FHA) antibodies than the combined DTaP–IPV–Hib and separate HBV vaccinations group (anti-PT: SMD = 0.60; 95% CI: 0.45, 0.75; *p* < 0.0001; anti-FHA: SMD = 0.40; 95% CI: 0.01, 0.78; *p* = 0.042). The levels of anti-pertactin (PRN) antibodies were lower in the combined DTaP–IPV–Hib vaccine group than in the combined DTaP–IPV and separate Hib vaccinations group (SMD = −0.13; 95% CI: −0.27, −0.00; *p* = 0.047). The individuals injected with the DTaP–HBV–IPV–Hib vaccine had a lower risk of pain and swelling than those injected with the combined DTaP–HBV–IPV and separate Hib vaccines (pain: RR = 0.79; 95% CI: 0.69, 0.91; *p* = 0.001; swelling: RR = 0.87; 95% CI: 0.78, 0.98; *p* = 0.020). However, the group that received the DTaP–HBV–IPV–Hib vaccine had a higher risk of fever than the group that received DTaP–HBV–IPV and separate Hib vaccinations (RR = 1.13; 95% CI: 1.02, 1.26; *p* = 0.021). Conclusions: This meta-analysis suggests that the combined vaccines (DTaP–IPV–Hib, DTaP–HBV–Hib, DTaP–HBV–IPV–Hib) are safe, well-tolerated, and provide immunogenic alternatives to separate vaccines in children. The combined DTaP–HBV–IPV–Hib vaccine showed a higher incidence of fever, which was lower than the cumulative incidence of fever induced by all vaccines. Future studies should evaluate the cost-effectiveness of using combined vaccines and compare the potency of different formulations to improve routine local or national childhood immunization programs.

## 1. Introduction

Emerging and re-emerging infectious diseases represent a significant threat to global economies and human health [1]. There is a growing awareness that vaccination is the most efficient and cost-effective measure for preventing and controlling infectious diseases [2,3]. With an increasing number of new vaccines, the success of vaccination programs and the introduction of new vaccines into already complex pediatric vaccination schedules can be a huge challenge globally [4]. Combination vaccines can be produced by grouping several antigens into one injection, representing an effective and important way of improving vaccination rates. Combination vaccines have been divided into two types by the World Health Organization: (1) vaccines containing several antigens in a single preparation and protecting against more than one disease and (2) those containing different strains or serotypes of the same organism, preventing one infectious disease [5]. Childhood combination vaccines have been widely used since the 1940s [6]. At present, five combination vaccines are licensed globally and widely used. The vaccine against pertussis, diphtheria, and tetanus, with combined diphtheria and tetanus toxoids and whole-cell pertussis (DTP), is one such vaccine that was recommended for infants and young children between the 1940s and 1990s. By 2020, the vaccination coverage rate of the third dose of the diphtheria–tetanus–pertussis (DTP_3_) vaccine was 74% [7]. The combined pentavalent vaccine against diphtheria–tetanus–acellular pertussis–hepatitis B–inactivated poliomyelitis (DTaP–HBV–IPV) and the diphtheria–tetanus–acellular pertussis–inactivated poliomyelitis and *Haemophilus influenzae* type B conjugate vaccine (DTaP–IPV–Hib) were licensed in 2002 and 2008, respectively [8]. The combined hexavalent diphtheria–tetanus-acellular pertussi–hepatitis B–inactivated poliomyelitis and *H. influenzae* type B conjugated vaccine (DTaP–HBV–IPV–Hib) was licensed in 2010 as the primary vaccination for infants and for use as a booster dose in the second year of life [9]. 

Uptake of vaccination may be affected by cost, compliance, convenience, and general beliefs about vaccine safety and effectiveness. Combination vaccines have several advantages over individual formulations, including reduced number of clinic visits and injections [10], decreased duration of infant distress, increased parental willingness to vaccinate, reduced operational and stocking costs, and increased vaccine-preventable disease coverage rates [10,11,12]. Among a varied panel of determinants of adherence to vaccination schedules, parental concerns about childhood vaccine safety and effectiveness are the most frequently cited factors influencing childhood vaccine uptake [13,14]. Many randomized controlled trials (RCT) and observational studies have been conducted to evaluate vaccine effectiveness and safety [15,16,17]. One Korean study showed that pentavalent DTaP–IPV–Hib vaccine immunogenicity was non-inferior to that of DTaP–IPV and Hib vaccines administered separately [18]. Meanwhile, RCTs comparing the effectiveness of the DTaP–HBV–Hib vaccine with those of the DTaP–HBV and Hib vaccines reported inconsistent results. A meta-analysis evaluated the efficacy of the the DTaP–HBV–Hib vaccine; however, it is unclear whether any of the combination vaccines is superior to the separate injections [19]. Therefore, the priority given to combination vaccines remains controversial.

The safety of combined vaccines is of primary importance because of their widespread use among healthy children as compared to separate vaccines. Two authorities have released statements regarding safety concerns about combined vaccines. WHO states: ‘In general, licensed combination vaccines are just as safe and effective as the single-disease vaccines’ [20]. The U.S. CDC states: ‘Before a combination vaccine is approved for use, it goes through careful testing to make sure the combination vaccine is as safe and effective as each of the individual vaccines given separately’ [10]. However, in contrast to the WHO and U.S. CDC statements, some studies have consistently shown that combined vaccines are not always as safe and effective as the individual vaccines. A study in *The Lancet* noted that the risk of the febrile seizures is increased following administration of DTP-based combined vaccines [21]. Similarly, a large Danish cohort study also showed an increased risk of febrile seizures following DTaP–IPV–Hib vaccination [22]. Two different studies in Germany and Italy have revealed that the sudden unexpected death rate was increased after application of the combined DTaP–HBV–Hib–IPV vaccine [23,24]. Therefore, it is necessary to carry out a systematic review and meta-analysis to fully evaluate the efficacy and safety of combined vaccines. 

To our knowledge, although combination vaccines are widely used globally, few comprehensive systematic reviews and meta-analyses have been published on their effectiveness and safety compared to those for separate injections or other appropriate comparators. A systematic review of the studies related to the immunogenicity and safety of the combined DTP–HBV–Hib vaccine versus separately administered DTP–HBV and Hib vaccines, which did not include the combined DTaP–IPV–Hib vaccine or the combined DTaP–IPV–HBV–Hib vaccine, was published in 2009 [19]. To date, an increasing number of randomized controlled trials have been published on combined DTaP–IPV–Hib, DTaP–HBV–Hib, and DTaP–IPV–HBV–Hib vaccines. The present systematic review and meta-analysis aimed to summarize the available evidence on the effectiveness and safety of three combination vaccines in children.

## 2. Methods

### 2.1. Literature Search

This systematic review and meta-analysis was performed according to the Preferred Reporting Items for Systematic Reviews and Meta-Analyses (PRISMA) [25]. We systematically searched electronic databases (MEDLINE, Embase, Cochrane, and Scopus) for RCTs on outcomes of the combined DTaP–IPV–Hib, DTaP–HBV–IPV, or DTaP–HBV–IPV–Hib vaccines compared to those of individual vaccines in infants. The outcomes of interest were vaccination immunogenicity and safety profiles. A systematic retrieval of literature from inception to 20 August 2021 was conducted. The search strategy involved the following keywords: “combined vaccines”. Figure 1 illustrates the flow diagram of the study selection process.

### 2.2. Selection Criteria

The inclusion of studies was based on the following criteria: (1) RCTs (including open-label or double-blind study design) comparing outcomes of vaccination with combined DTaP–IPV–Hib, DTaP–HBV–IPV, or DTaP–HBV–IPV–Hib with those of separate vaccines or placebo; (2) vaccines administered to children aged ≤2 years; and (3) reported measures of immunogenicity (non-inferiority of geometric mean titers (GMTs)) or safety (local and systemic adverse events (AEs)). The exclusion criteria for studies were as follows: (1) non-original studies: review, meta-analysis, systematic review; standards, guidelines, teaching materials, books, or conference abstracts; (2) non-RCT studies, including case reports, case studies, case series studies, and other article types; (3) non-human studies; (4) outcomes of interest not reported; and (5) no full text available or duplicate publications. 

### 2.3. Data Extraction and Quality Assessment

Two investigators (HJM and JTX) independently screened and reviewed the shortlisted articles (and any Appendix A) and extracted the relevant information. Any discrepancies were discussed and resolved by a third reviewer (BL). Reference lists of the retrieved articles were reviewed to identify any additional potentially eligible studies. Additional articles were manually retrieved from the journal’s official website. 

Data on the following variables were extracted from each eligible study: first author name, publication year, study design, country, geographic location, participant age and sex, vaccine type, immunogenicity parameters (including anti-diphtheria, anti-tetanus, anti-hepatitis B, anti-pertussis toxoid (PT), anti-filamentous hemagglutinin (FHA), anti- pertactin (PRN), anti-polyribosylribitol phosphate (PRP), polio serotype 1, polio serotype 2, and polio serotype 3), and incidence rates of local and systemic reactions (pain, redness, swelling, fever, irritability, loss of appetite, restlessness, sleepiness, unusual drowsiness, vomiting).

We used the Cochrane ROB 2.0 tool for systematic reviews of intervention studies to assess the quality of the included studies by evaluating six ROB items. The six domains were as follows: sequence generation, allocation concealment, blinding of participants and personnel, blinding of outcome assessments, incomplete outcome data, and selective outcome reporting. The categories ‘low’, ‘high’, and ‘unclear’ risk of bias were used to assess each category. 

### 2.4. Statistical Analysis

We estimated the summary standardized mean differences (SMDs, Cohen’s d) with 95% confidence intervals (CIs) for continuous data using pairwise meta-analysis. The relative risks (RRs) with 95% CIs were used to calculate the reactogenicity (adverse events) in the vaccinated groups compared with that in the control groups. The weight (%) of each study was assigned based on the inverse of the variance. Greater weights had a greater impact on the combined results. Chi-square statistics and *I*^2^ test values were used to assess the heterogeneity of all studies. The random effects model was used if statistical differences in contexts of high heterogeneity existed (*p*-values of <0.10 or *I*^2^ of >50%). Otherwise, a fixed-effects model was used. Subgroup analyses of different vaccine types and geographic locations were performed to explore the potential impact of study characteristics on the pooled effect size. For comparisons with more than 10 original studies, publication bias was assessed using a funnel plot with Begg’s test and Egger’s test. All data analyses were performed using Stata (version 15.0, Stata Corp LP, College Station, TX, USA). 

## 3. Results

We identified 3626 potentially eligible articles in the MEDLINE, Embase, Cochrane, and Scopus databases. After removing duplicate records, 2913 records remained. Most articles (*n* = 2888) did not meet the inclusion criteria and were excluded after reviewing their titles, abstracts, and full text. Finally, 25 articles were included in the meta-analysis (Figure 1). We identified 10 studies reporting on the DTaP–IPV–Hib vaccine, 8 studies reporting on the DTaP–HBV–Hib vaccine, and 7 studies reporting on the DTaP–HBV–IPV–Hib vaccine. The total number of subjects included in the meta-analysis was 2690 in the combined vaccine group and 2398 in the control group. 

### 3.1. Study Characteristics

The basic characteristics of the 25 included studies are presented in Table 1. The RCTs were conducted in Europe (n = 11), Asia (n = 9), America (n = 2), Africa (n = 1), and Oceania (n = 1). The age of the participants ranged from 6 to 20 weeks. Only three of the included studies had a double-blind design, while the other studies were open-label RCTs. Overall, 18 studies received industry funding; among them, nine studies received funding from GlaxoSmithKline. Three studies did not receive any industry funding, and the other studies did not report whether they received financial support from the industry. In Klein et al.’s study, the combined DTaP–HBV–IPV–Hib vaccine had two control comparators (i.e., DTaP–HBV–IPV+Hib and DTaP–HBV–IPV+HBV), which we considered as separate records in the analysis.

### 3.2. Pooled Immunogenicity of Combined Vaccine

The immunogenicity (including anti-diphtheria, anti-tetanus, anti-hepatitis B, anti-pertussis (PT), anti-filamentous hemagglutinin (FHA), anti-pertactin (PRN), anti-polyribosyl ribitol phosphate (PRP), polio serotype 1, polio serotype 2, and polio serotype 3) values of different types of combined vaccines are shown in Table 2 and Appendix A. The acellular pertussis (aP) antigens include purified pertussis toxin (PT), filamentous haemagglutinin (FHA), and pertactin (PRN). Due to the high heterogeneity among the immunogenicity parameters (*I*^2^ > 50% or *p* < 0.05), random effects models were used to calculate pooled estimates. There were differences among the combined and separate vaccines with respect to the parameters associated with anti-tetanus (SMD = −0.15; 95% CI: −0.26, −0.04; *p* = 0.006) and anti-PRP (SMD = −0.53; 95% CI: −0.79, −0.27; *p* < 0.001) responses when all combined vaccines were analyzed together. The combined and separate vaccines did not differ with respect to the other immunogenicity parameters. 

Among combined vaccine groups, the DTaP–HBV–Hib group had lower levels of anti-tetanus antibodies than the DTaP–HBV+Hib group (SMD = −0.23; 95% CI: −0.42, −0.05; *p* = 0.013). The DTaP–HBV–IPV–Hib group had higher levels of anti-PT and anti-FHA antibodies than the DTaP–IPV–Hib+HBV group (anti-PT: SMD = 0.60; 95% CI: 0.45, 0.75; *p* < 0.0001; anti-FHA: SMD = 0.40; 95% CI: 0.01, 0.78; *p* = 0.042). The levels of anti-PRN antibodies were lower in the DTaP–IPV–Hib group than in the DTaP–IPV+Hib group (SMD = −0.13; 95% CI: −0.27, −0.00; *p* = 0.047). Moreover, the levels of anti-PRP antibodies were lower in the DTaP–IPV–Hib group than in the DTaP–IPV+Hib group (SMD = −0.83; 95% CI: −1.44, −0.22; *p* = 0.007); they were also lower in the DTaP–HBV–Hib group than in the DTaP–HBV+Hib group (SMD = −0.37; 95% CI: −0.73, −0.01; *p* = 0.043), and in the DTaP–HBV–IPV–Hib group than in the DTaP–IPV–Hib+HBV group (SMD = −0.61; 95% CI: −1.03, −0.69; *p* = 0.005). 

### 3.3. Pooled Acceptability of Combined Vaccines

The incidence rates of local (pain, redness, and swelling) and systemic reactions (diarrhea, fever, irritability, loss of appetite, restlessness, sleepiness, unusual drowsiness, vomiting) associated with different types of combined vaccines are shown in Table 3 and Appendix A. The incidence rates of side effects were comparable among all combined vaccines when they were analyzed together. Meanwhile, the individuals injected with DTaP–HBV–IPV–Hib had a lower risk of pain and swelling than those injected with DTaP–HBV–IPV+Hib (pain: RR = 0.79; 95% CI: 0.69, 0.91; *p* = 0.001; swelling: RR = 0.87; 95% CI: 0.78, 0.98; *p* = 0.020). In addition, the group that received DTaP–HBV–IPV–Hib had a higher risk of fever than the group that received DTaP–HBV–IPV+Hib (RR = 1.13; 95% CI: 1.02, 1.26; *p* = 0.021) or DTaP–IPV–Hib+HBV (RR = 1.26; 95% CI: 1.08, 1.47; *p* = 0.003). Moreover, the group that received DTaP–HBV–IPV–Hib had a higher risk of irritability than the group that received DTaP–IPV–Hib+HBV (RR = 1.18; 95% CI: 1.04, 1.33; *p* = 0.010).

### 3.4. Risk of Bias and Quality Assessment

We performed Egger’s test and Begg’s test to evaluate publication bias (Appendix A). The publication bias test was performed when there were more than 10 studies in a given category. The funnel plots revealed that there was a low risk of publication bias among the studies assessing the immunogenicity and reactogenicity of combined vaccines (all *p* > 0.05 for Egger’s and Begg’s tests).

As shown in Figure 2, in all studies, the risk of bias associated with random sequence generation was low. Meanwhile, 11, 20, 8, 3, and 20 studies had a high risk of bias associated with allocation concealment, blinding of participants and personnel, blinding of outcome assessment, incomplete outcome data, and selective reporting, respectively. 

## 4. Discussion

Combination vaccines can improve immunization coverage and reduce the number of medical visits, improve the quality of healthcare services and reduce the overall costs for healthcare systems and society [26,27]. However, because of the lack of large sample-size and multi-population clinical reports, the safety and efficiency of combined vaccination still remains controversial. This study provides a summary of the safety and effectiveness of childhood combination vaccines against clinical outcomes that are relevant for decision makers. Three currently available combination vaccines for childhood immunization were evaluated in our study: the combined DTaP–IPV–Hib vaccine, the combined DTaP–HBV–Hib vaccine, and the combined DTaP–HBV–IPV–Hib vaccine. 

Differences in immunogenicity were found among combined vaccines and their separate comparators for anti-tetanus immunity (SMD = −0.15; 95% CI: −0.26, −0.04; *p* = 0.006) and anti-PRP immunity (SMD = −0.53; 95% CI: −0.79, −0.27; *p* < 0.001) when all combined vaccines were analyzed together. These results are consistent with those of a meta-analysis comparing the combined DTP–HBV–Hib vaccine and separately administered DTP–HBV and Hib vaccines for the primary prevention of diphtheria, tetanus, pertussis, hepatitis B, and *H. influenzae* B [19]. In addition, the group that received the DTaP–HBV–Hib combination vaccine had lower levels of anti-tetanus antibodies than the group that received the DTaP–HBV+Hib combination vaccine (SMD = −0.23; 95% CI: −0.42, −0.05; *p* = 0.013). With a consistent finding, a similar study found that preexisting immune responses to carrier proteins significantly lessen the production of anti-tetanus and Hib antibodies [28]. Similarly, the levels of anti-PRP antibodies were lower in the DTaP–IPV/Hib group than in the DTaP–IPV+Hib group (SMD = −0.83; 95% CI: −1.44, −0.22; *p* = 0.007); they were also lower in the DTaP–HBV–Hib group than in the DTaP–HBV+Hib group (SMD = −0.37; 95% CI: −0.73, −0.01; *p* = 0.043), and in the DTaP–HBV–IPV–Hib group than in the DTaP–IPV–Hib+HBV group (SMD = −0.61; 95% CI: −1.03, −0.69; *p* = 0.005). Similar findings were made in other studies [29,30,31,32]. Anti-PRP induces a lower immune response when combined with acellular pertussis antigens than when it is administered as a separate injection [33,34,35]. The potential mechanism responsible for the immunogenicity differences among the combined and separate vaccines may be associated with the complex physical or chemical intermolecular interactions between the mixture component of combined vaccination [36]. Typically, the main characteristic feature of the DTaP vaccines is that vaccine antigens are adsorbed to adjuvants. Another possible reason may be other normally absorbed bioactive components of combined vaccines being displaced from adjuvants [37,38]. In contrast, the levels of anti-PT and anti-FHA antibodies associated with DTaP–HBV–IPV–Hib were higher than those associated with DTaP–IPV–Hib+HBV (anti-PT: SMD = 0.60; 95% CI: 0.45, 0.75; *p* < 0.0001; anti-FHA: SMD = 0.40; 95% CI: 0.01, 0.78; *p* = 0.042). Vaccines that contain aP can contain PT, FHA, PRN, and fimbriae types 2 and 3 [39]. A previous review has shown that aP-containing vaccines with three or more components had higher efficacy against typical whooping cough than those containing one or two components [40]. These findings suggest that the recipients of the DTaP–HBV–IPV–Hib vaccine were relatively better protected against whooping cough, although the pertussis virus titers were equivalent in the DTaP–HBV–IPV–Hib and DTaP–IPV–Hib vaccines. 

In the present study, reactogenicity (adverse events) values were similar among all vaccine types. However, the rates of pain and swelling differed between the DTaP–HBV–IPV–Hib combination vaccine and separate vaccines (pain: RR = 0.79, 95% CI: 0.69, 0.91; swelling: RR = 0.87, 95% CI: 0.78, 0.98). These safety results are consistent with those of previous studies that used the same vaccine [41,42,43]. This finding may be accounted for by the fact that additional injections can be avoided with the use of combination vaccines [44]. The group that received DTaP–HBV–IPV–Hib had a higher risk of fever than the group that received DTaP–HBV–IPV+Hib (RR = 1.13; 95% CI: 1.02, 1.26; *p* = 0.021) or DTaP–IPV–Hib+HBV (RR = 1.26; 95% CI: 1.08, 1.47; *p* = 0.003). The results were consistent with those of a previously reported statistical model, which indicated that fever was the most common immediate systemic reaction found in children receiving vaccines with DTaP antigens [45]. However, many RCTs have shown little difference in the incidence of fever among DTaP–HBV–IPV–Hib combined and separate vaccines, which is in contrast to the present results. These discrepancies may be accounted for by the differences in study populations, sample sizes, and clinical heterogeneity among studies. Irritability was the most frequently reported solicited systemic reaction. The meta-analysis revealed that the group that received DTaP–HBV–IPV–Hib had a higher risk of irritability than the group that received DTaP–IPV–Hib+HBV (RR = 1.18; 95% CI: 1.04, 1.33; *p* = 0.010).

This meta-analysis included 5088 healthy children from different countries and ethnic groups. To minimize the risk of selection bias and provide more adequate and reliable evidence, we included only RCTs. However, this study had some limitations. First, in the present study, many randomized trials did not report the specifics of random sequence generation, allocation concealment, or blinding. Some blinding was only applied to serum analysis by laboratory technicians. Second, although vaccination schedules were similar among the included studies, immunogenicity was measured at different time points. We included immunogenicity data after the third vaccination, while the immunogenicity profile might differ after booster vaccination. Third, reporting of localized reactions (pain, redness, and swelling) was different among the included studies. Some studies reported localized reactions separately for combined and separate vaccines, while other studies reported an overall incidence. When the locations are double-counted in separate vaccines, reports of adverse events may favor a combined vaccine. Fourth, the differences in study locations and healthcare environments may have biased the presented estimates. Fifth, the present study did not include any unpublished research reports, non-English articles, or dissertations; therefore, potentially relevant studies may have been excluded from this systematic review, resulting in some publication bias affecting the presented estimates.

We used the Cochrane ROB 2.0 tool to assess the quality of studies based on six ROB items. In all studies, the risk of bias associated with random sequence generation was low. Meanwhile, 11, 20, 8, 3, and 20 studies had a high risk of bias associated with allocation concealment, blinding of participants and personnel, blinding of outcome assessment, incomplete outcome data, and selective reporting, respectively. The differences in the risk of bias were mostly determined by one study per category; it is not clear whether the results can be generalized to all vaccines. The present findings should be approached and interpreted with caution, as high-quality data are lacking.

In conclusion, the present findings suggest that the combined vaccines (DTaP–IPV–Hib, DTaP–HBV–Hib, DTaP–HBV–IPV–Hib) are safe, well-tolerated, and provide immunogenic alternatives to separate vaccines in children. The combined DTaP–HBV–IPV–Hib vaccine showed relatively high incidence rates of fever; however, these rates remained lower than the cumulative incidence rates of fever associated with all vaccines. The present findings suggest that healthcare providers should present parents with the risks and benefits of both vaccination options before the administration of combined vaccines. The vaccination schedules among separate vaccines in children potentially overlapped, so the combined vaccines would be convenient and achieve favorable cost-effectiveness [46,47]. Recently, a study from Malaysia suggested that the use of a hexavalent DTaP–HBV–IPV–Hib combination vaccine had a lower cost per dose and demonstrated substantial direct and indirect cost savings for healthcare providers and parents [48]. A study from China in 2019 reported savings of 2.2 billion RMB for families in Guangdong province as a result of the co-administration of multiple vaccines [49]. Future studies should evaluate the cost-effectiveness of using combined vaccines and compare the potency of different formulations to improve national childhood immunization programs.

## Figures and Tables

**Figure 1 vaccines-10-00472-f001:**
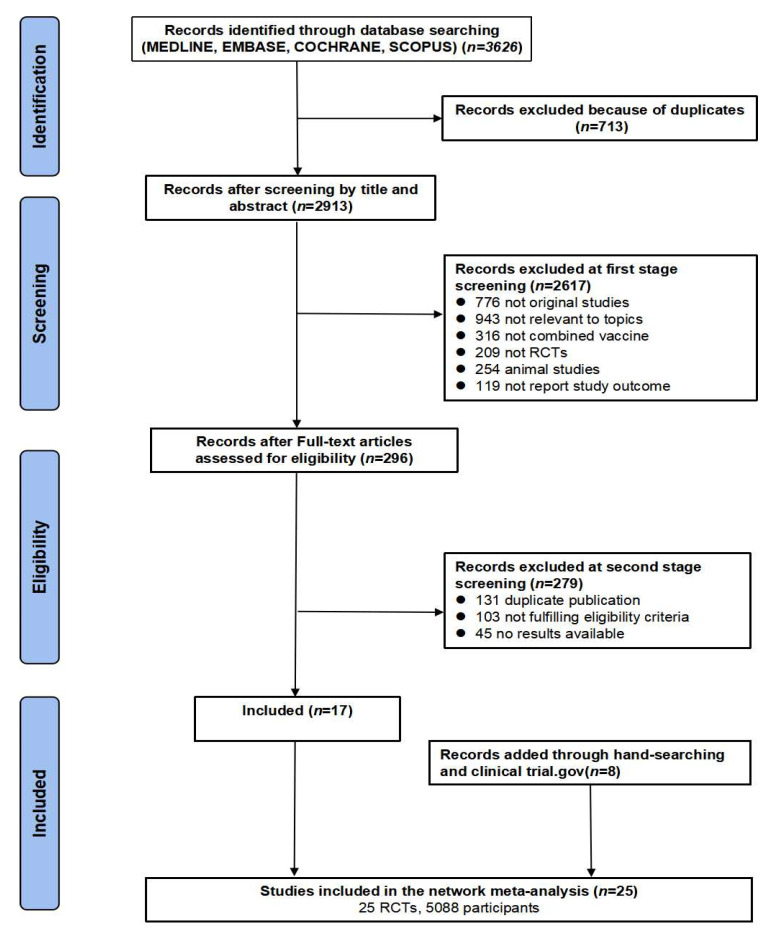
PRISMA flow diagram of study selection process.

**Figure 2 vaccines-10-00472-f002:**
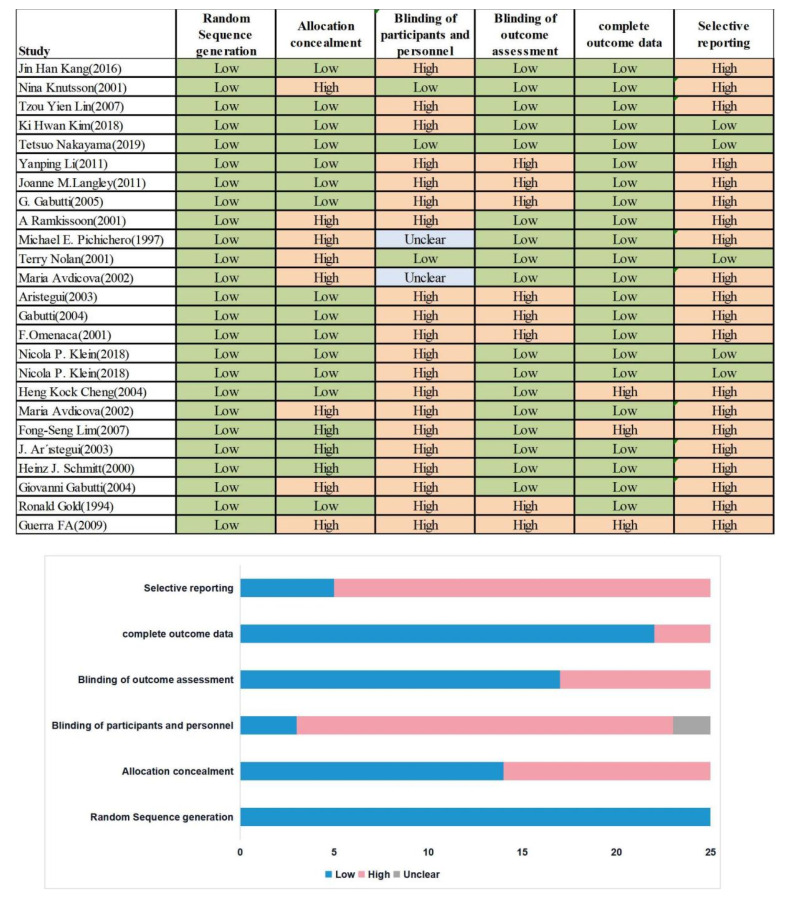
Quality Assessment of included studies.

**Table 1 vaccines-10-00472-t001:** Basic Characteristics of included studies.

ID	First Author	Year	Country	Study Design	Age Range	Vaccine Comparisons	Company Funding
1	Jin Han Kang	2016	Korea	Open-label, randomized, and controlled trial	1.8–2.3 months	DTaP–IPV–Hib vs. DTaP–IPV+Hib	Yes (Sanofi PasteurSA, Lyon, France)
2	Nina Knutsson	2001	Swedish	Randomized, a double-blind placebo-controlled efficacy trial	_	DTaP–IPV–Hib vs. DTaP–IPV+Hib	Yes (North American Vaccine Inc., Maryland, USA)
3	Tzou Yien Lin	2007	China	Open-label, randomized, controlled trial	8 weeks	DTaP–IPV–Hib vs. DTaP–IPV+Hib	Yes (Sanofi Pasteur, Lyon, France)
4	Ki Hwan Kim	2018	Korea	A Phase III, open-label, randomized, controlled trial	42–69 days	DTaP–IPV–Hib vs. DTaP–IPV+Hib	Yes (GlaxoSmithKline Biologicals SA)
5	Tetsuo Nakayama	2019	Japan	A Phase III, modified double-blind, active-controlled, 2-arm, balanced trial	_	DTaP–IPV–Hib vs. DTaP–IPV+Hib	Yes
6	Yanping Li	2011	China	A Phase III, open-label, randomized, controlled trial	_	DTaP–IPV–Hib vs. DTaP–IPV+Hib	Yes (GlaxoSmithKline Biologicals SA)
7	Joanne M.Langley	2011	Canada	Randomized controlled trial	_	DTaP–IPV–Hib vs. DTaP–IPV+Hib	No data
8	Ronald Gold	1994	Canada	Randomized controlled trial	_	DTaP–IPV–Hib vs. DTaP–IPV+Hib	No data
9	Guerra FA	2009	US	Randomized controlled trial	_	DTaP–IPV–Hib vs. DTaP–IPV+Hib	_
10	G. Gabutti	2005	Italy	Open, randomized, multicentre	12–16 weeks	DTaP–HBV–Hib vs. DTaP–HBV+Hib	Yes (GSK Biologicals, Rixensart, Belgium)
11	A Ramkissoon	2001	South Africa	Open, randomized comparative trial	_	DTaP–HBV–Hib vs. DTaP–HBV+Hib	No
12	Michael E. Pichichero	1997	UK	A multicenter, prospective, randomized trial	6–12 weeks	DTaP–HBV–Hib vs. DTaP–HBV+Hib	No
13	Terry Nolan	2001	Melbourne	A randomised double-blind trial	_	DTaP–HBV–Hib vs. DTaP–HBV+Hib	Yes
14	Maria Avdicova	2002	Slovak	Open-label, randomized, controlled trial	8–20 weeks	DTaP–HBV–Hib vs. DTaP–HBV+Hib	Yes (GlaxoSmithKline Biologicals, Rixensart, Belgium)
15	Aristegui	2003	Spain	Open randomized, comparative Phase III multicenter trial	_	DTaP–HBV–Hib vs. DTaP–HBV+Hib	Yes (GSK Biologicals, Rixensart, Belgium)
16	Gabutti	2004	Germany and Italy	Open, Phase III, randomized trial	12–16 weeks	DTaP–HBV–Hib vs. DTaP–HBV+Hib	Yes (GSK Biologicals, Rixensart, Belgium)
17	F.Omenaca	2001	Greece, Spain, and Switzerland	Open, Phase III, randomized trial	8–12 weeks	DTaP–HBV–Hib vs. DTaP–HBV+Hib	Yes
18	Nicola P. Klein	2018	US	Open-label, randomized, controlled trial	6–12 weeks	DTaP–HBV–IPV–Hib vs. DTaP–HBV–IPV+Hib	Yes (GlaxoSmithKline Biologicals S.A)
19	Nicola P. Klein	2018	US	Open-label, randomized, controlled trial	6–12 weeks	DTaP–HBV–IPV–Hib vs. DTaP–HBV–IPV+HBV	Yes (GlaxoSmithKline Biologicals S.A)
20	Heng Kock Cheng	2004	Singapore	Open-label, randomized, controlled trial	_	DTaP–HBV–IPV–Hib vs. DTaP–HBV–IPV+HBV	_
21	Maria Avdicova	2002	Slovakia	Open-label, randomized, controlled trial	8–20 weeks	DTaP–HBV–IPV–Hib vs. DTaP–HBV–IPV+HBV	Yes (GlaxoSmithKline Biologicals)
22	Fong-Seng Lim	2007	Singapore	Open-label, randomized, controlled trial	12–16 weeks	DTaP–HBV–IPV–Hib vs. DTaP–HBV–IPV+HBV	_
23	J. Ar’ıstegui	2003	Spain	An open, randomized, multi-center, comparative Phase IIIb clinical trial	8–11 weeks	DTaP–HBV–IPV–Hib vs. DTaP–HBV–IPV+HBV	Yes (GlaxoSmithKline Biologicals S.A)
24	Heinz J. Schmitt	2000	Germany	An open, randomized, multi-center trial	8–16 weeks	DTaP–HBV–IPV–Hib vs. DTaP–HBV–IPV+Hib	No
25	Giovanni Gabutti	2004	Germany and Italy	An open, randomized, multi-center trial	_	DTaP–HBV–IPV–Hib vs. DTaP–HBV–IPV+Hib	Yes (GSK Biologicals, Rixensart, Belgium)

**Table 2 vaccines-10-00472-t002:** The meta-analysis of immunogenicity of different types of combined vaccines.

Variables	Vaccine Group	No. of Studies	SMD (95% CI)/GMTs	% Weight	z	*p*-Effect	*I* ^2^	*p*-Heterogeneity
Anti-diphtheria								
	DTaP–IPV–Hib vs. DTaP–IPV+Hib	8	0 (−0.08, 0.07)	48.18	−0.109	0.913	27.80%	0.206
	DTaP–HBV–Hib vs. DTaP–HBV+Hib	7	−0.08 (−0.23, 0.06)	27.61	−1.115	0.265	41.90%	0.112
	DTaP–HBV–IPV–Hib vs. DTaP–HBV–IPV+Hib	2	0.03 (−0.13, 0.18)	11.14	0.336	0.737	0.00%	0.326
	DTaP–HBV–IPV–Hib vs. DTaP–IPV–Hib+HBV	3	0.19 (−0.13, 0.51)	13.06	1.186	0.236	74.50%	0.020
	Overall	20	0.01 (−0.07, 0.08)	100	0.124	0.902	53.50%	0.003
Anti-tetanus								
	DTaP–IPV–Hib vs. DTaP–IPV+Hib	8	−0.12 (−0.3, 0.05)	44.79	−1.396	0.163	85.5%	<0.001
	DTaP–HBV–Hib vs. DTaP–HBV+Hib	7	−0.23 (−0.42, −0.05)	30.45	−2.487	0.013	63.7%	0.011
	DTaP–HBV–IPV–Hib vs. DTaP–HBV–IPV+Hib	2	−0.2 (−0.41, 0.02)	10.90	−1.790	0.073	49.3%	0.160
	DTaP–HBV–IPV–Hib vs. DTaP–IPV–Hib+HBV	3	−0.02 (−0.37, 0.34)	13.86	−0.082	0.935	79.5%	0.008
	Overall	20	−0.15 (−0.26, −0.04)	100.00	−2.737	0.006	77.7%	<0.001
Anti-hepatitis B								
	DTaP–HBV–Hib vs. DTaP–HBV+Hib	8	−0.21 (−0.44, 0.02)	60.55	−1.817	0.069	82.2%	<0.001
	DTaP–HBV–IPV–Hib vs. DTaP–HBV–IPV+Hib	2	−0.02 (−0.4, 0.36)	16.62	−0.091	0.928	83.7%	0.013
	DTaP–HBV–IPV–Hib vs. DTaP–IPV–Hib+HBV	3	0.15 (−0.33, 0.63)	22.83	0.619	0.536	88.7%	<0.001
	Overall	13	−0.09 (−0.31, 0.12)	100.00	−0.870	0.385	88.9%	<0.001
Anti-pertussis								
	DTaP–IPV–Hib vs. DTaP–IPV+Hib	8	0.27 (−0.16, 0.69)	43.57	1.240	0.215	97.5%	<0.001
	DTaP–HBV–Hib vs. DTaP–HBV+Hib	6	0.07 (−0.22, 0.35)	30.13	0.444	0.657	80.1%	<0.001
	DTaP–HBV–IPV–Hib vs. DTaP–HBV–IPV+Hib	2	−0.15 (−0.3, 0.01)	10.86	−1.889	0.059	0.0%	0.999
	DTaP–HBV–IPV–Hib vs. DTaP–IPV–Hib+HBV	3	0.6 (0.45, 0.75)	15.45	7.962	0.000	0.0%	0.450
	Overall	19	0.21 (−0.03, 0.44)	100.00	1.735	0.083	95.2%	<0.001
Anti-FHA								
	DTaP–IPV–Hib vs. DTaP–IPV+Hib	6	0.28 (−0.2, 0.76)	38.86	1.153	0.249	96.7%	<0.001
	DTaP–HBV–Hib vs. DTaP–HBV+Hib	5	−0.08 (−0.3, 0.15)	29.83	−0.640	0.522	65.3%	0.021
	DTaP–HBV–IPV–Hib vs. DTaP–HBV–IPV+Hib	2	−0.13 (−0.3, 0.04)	12.98	−1.464	0.143	22.3%	0.257
	DTaP–HBV–IPV–Hib vs. DTaP–IPV–Hib+HBV	3	0.4 (0.01, 0.78)	18.33	2.034	0.042	82.4%	0.003
	Overall	16	0.14 (−0.09, 0.38)	100.00	1.183	0.237	93.5%	<0.001
Anti-PRN								
	DTaP–IPV–Hib vs. DTaP–IPV+Hib	3	−0.13 (−0.27, 0)	24.66	−1.984	0.047	0.0%	0.565
	DTaP–HBV–Hib vs. DTaP–HBV+Hib	5	−0.2 (−0.53, 0.13)	35.85	−1.200	0.230	83.4%	<0.001
	DTaP–HBV–IPV–Hib vs. DTaP–HBV–IPV+Hib	2	0.08 (−0.19, 0.35)	16.92	0.566	0.571	67.7%	0.078
	DTaP–HBV–IPV–Hib vs. DTaP–IPV–Hib+HBV	3	0.08 (−0.5, 0.66)	22.58	0.281	0.778	92.4%	<0.001
	Overall	13	−0.07 (−0.25, 0.1)	100.00	−0.806	0.420	83.6%	<0.001
Anti-PRP								
	DTaP–IPV–Hib vs. DTaP–IPV+Hib	6	−0.83 (−1.44, −0.22)	34.06	−2.680	0.007	97.6%	<0.001
	DTaP–HBV–Hib vs. DTaP–HBV+Hib	7	−0.37 (−0.73, −0.01)	37.92	−2.029	0.043	91.2%	<0.001
	DTaP–HBV–IPV–Hib vs. DTaP–HBV–IPV+Hib	2	−0.61 (−1.03, −0.19)	11.50	−2.828	0.005	86.2%	0.007
	DTaP–HBV–IPV–Hib vs. DTaP–IPV–Hib+HBV	3	−0.24 (−0.69, 0.22)	16.52	−1.003	0.316	87.8%	<0.001
	Overall	18	−0.53 (−0.79, −0.27)	100.00	−4.005	0.000	94.9%	<0.001
Polio serotype 1								
	DTaP–IPV–Hib vs. DTaP–IPV+Hib	7	0.06 (−0.19, 0.3)	49.24	0.474	0.636	89.6%	<0.001
	DTaP–HBV–Hib vs. DTaP–HBV+Hib	4	−0.04 (−0.18, 0.1)	25.25	−0.539	0.590	14.2%	0.321
	DTaP–HBV–IPV–Hib vs. DTaP–HBV–IPV+Hib	2	0.24 (−0.39, 0.87)	14.04	0.747	0.455	94.0%	<0.001
	DTaP–HBV–IPV–Hib vs. DTaP–IPV–Hib+HBV	2	−0.09 (−0.3, 0.12)	11.46	−0.805	0.421	0.0%	0.634
	Overall	15	0.04 (−0.12, 0.19)	100.00	0.444	0.657	84.4%	<0.001
Polio serotype 2								
	DTaP–IPV–Hib vs. DTaP–IPV+Hib	7	0.06 (−0.16, 0.27)	49.01	0.521	0.602	85.9%	<0.001
	DTaP–HBV–Hib vs. DTaP–HBV+Hib	4	−0.02 (−0.24, 0.19)	25.40	−0.205	0.838	59.9%	0.058
	DTaP–HBV–IPV–Hib vs. DTaP–HBV–IPV+Hib	2	0.4 (−0.41, 1.21)	13.95	0.961	0.336	96.3%	<0.001
	DTaP–HBV–IPV–Hib vs. DTaP–IPV–Hib+HBV	2	−0.12 (−0.33, 0.09)	11.64	−1.108	0.268	0.0%	0.701
	Overall	15	0.06 (−0.11, 0.22)	100.00	0.692	0.489	86.4%	<0.001
Polio serotype 3								
	DTaP–IPV–Hib vs. DTaP–IPV+Hib	7	0.03 (−0.22, 0.28)	48.69	0.253	0.801	89.7%	<0.001
	DTaP–HBV–Hib vs. DTaP–HBV+Hib	4	0.04 (−0.2, 0.27)	25.58	0.330	0.742	66.8%	0.029
	DTaP–HBV–IPV–Hib vs. DTaP–HBV–IPV+Hib	2	0.21 (−0.77, 1.2)	13.87	0.422	0.673	97.5%	<0.001
	DTaP–HBV–IPV–Hib vs. DTaP–IPV–Hib+HBV	2	−0.08 (−0.54, 0.38)	11.86	−0.328	0.743	68.6%	0.074
	Overall	15	0.05 (−0.13, 0.23)	100.00	0.543	0.587	88.6%	0.000

Immunogenicity: non-inferiority of genometric mean titers (GMTs): SMD/GMTs: standard mean difference of GMTs. Abbreviations: PT, pertussis; FHA, filamentous hemagglutinin; PRN, pertactin; PRP, polyribosyl ribitol phosphate; HBV, hepatitis B; DTaP, tetanus, diphtheria, and acellular pertussis vaccine.

**Table 3 vaccines-10-00472-t003:** The meta-analysis of local and systemic reactions of different types of combined vaccines.

Variables	Vaccine Group	No. of Studies	RR (95% CI)	% Weight	z	*p*-Effect	*I* ^2^	p-Heterogeneity
Redness								
	DTaP–IPV–Hib vs. DTaP–IPV+Hib	5	1.04 (0.87, 1.24)	28.24	0.398	0.691	67.00%	0.017
	DTaP–HBV–Hib vs. DTaP–HBV+Hib	7	0.92 (0.81, 1.06)	50.85	−1.129	0.259	88.70%	0.000
	DTaP–HBV–IPV–Hib vs. DTaP–HBV–IPV+Hib	2	0.80 (0.63, 1.03)	13.73	−1.742	0.082	75.20%	0.045
	DTaP–HBV–IPV–Hib vs. DTaP–IPV–Hib+HBV	1	1.17 (1.01, 1.36)	7.18	2.040	0.041	-	-
	Overall	15	0.95 (0.86, 1.05)	100.00	−0.982	0.326	86.80%	0.000
Pain								
	DTaP–IPV–Hib vs. DTaP–IPV+Hib	4	0.94 (0.86, 1.02)	24.35	−1.512	0.131	0.00%	0.574
	DTaP–HBV–Hib vs. DTaP–HBV+Hib	7	1.04 (0.94, 1.15)	49.9	0.673	0.501	81.10%	0.000
	DTaP–HBV–IPV–Hib vs. DTaP–HBV–IPV+Hib	2	0.80 (0.69, 0.91)	12.58	−3.239	0.001	9.70%	0.293
	DTaP–HBV–IPV–Hib vs. DTaP–IPV–Hib+HBV	2	0.76 (0.34, 1.73)	13.17	−0.656	0.512	97.60%	0.000
	Overall	15	0.94 (0.85, 1.04)	100.00	−1.151	0.25	87.60%	0.000
Swelling								
	DTaP–IPV–Hib vs. DTaP–IPV+Hib	5	1.07 (0.86, 1.33)	28.6	0.623	0.534	82.60%	0.000
	DTaP–HBV–Hib vs. DTaP–HBV+Hib	7	0.97 (0.85, 1.11)	46.68	−0.439	0.66	86.50%	0.000
	DTaP–HBV–IPV–Hib vs. DTaP–HBV–IPV+Hib	2	0.87 (0.78, 0.98)	12.57	−2.331	0.02	0.00%	0.402
	DTaP–HBV–IPV–Hib vs. DTaP–IPV–Hib+HBV	2	0.90 (0.52, 1.58)	12.15	−0.358	0.72	93.90%	0.000
	Overall	16	0.98 (0.89, 1.09)	100.00	−0.359	0.72	86.50%	0.000
Diarrhea								
	DTaP–IPV–Hib vs. DTaP–IPV+Hib	1	0.58 (1.2, 0)	35.49	−0.961	0.336	-	-
	DTaP–HBV–Hib vs. DTaP–HBV+Hib	2	0.95 (1.43, 0)	64.51	1.436	0.151	0.00%	0.321
	Overall	3	0.87 (1.26, 0)	100.00	0.499	0.618	47.90%	0.147
Fever								
	DTaP–IPV–Hib vs. DTaP–IPV+Hib	7	1.00 (0.92, 1.09)	37.79	0.051	0.959	17.60%	0.296
	DTaP–HBV–Hib vs. DTaP–HBV+Hib	6	0.99 (0.79, 1.24)	38.74	−0.104	0.917	95.20%	0.000
	DTaP–HBV–IPV–Hib vs. DTaP–HBV–IPV+Hib	2	1.13 (1.02, 1.26)	11.76	2.308	0.021	0.00%	0.397
	DTaP–HBV–IPV–Hib vs. DTaP–IPV–Hib+HBV	2	1.26 (1.08, 1.47)	11.71	2.985	0.003	20.30%	0.263
	Overall	17	1.03 (0.93, 1.15)	100.00	0.575	0.565	87.70%	0.000
Irritability								
	DTaP–IPV–Hib vs. DTaP–IPV+Hib	6	0.92 (0.85, 1)	36.15	−1.945	0.052	13.80%	0.326
	DTaP–HBV–Hib vs. DTaP–HBV+Hib	6	0.95 (0.83, 1.1)	46.28	−0.671	0.502	89.40%	0.000
	DTaP–HBV–IPV–Hib vs. DTaP–HBV–IPV+Hib	1	1.10 (0.85, 1.42)	4.95	0.716	0.474	-	-
	DTaP–HBV–IPV–Hib vs. DTaP–IPV–Hib+HBV	2	1.18 (1.04, 1.33)	12.62	2.587	0.010	0.00%	0.674
	Overall	15	0.97 (0.9, 1.06)	100.00	−0.692	0.489	78.60%	0.000
Loss of appetite								
	DTaP–IPV–Hib vs. DTaP–IPV+Hib	6	1.01 (0.93, 1.1)	30.75	0.247	0.805	0.00%	0.682
	DTaP–HBV–Hib vs. DTaP–HBV+Hib	4	1.03 (0.98, 1.09)	51.51	1.092	0.275	24.40%	0.265
	DTaP–HBV–IPV–Hib vs. DTaP–HBV–IPV+Hib	1	0.99 (0.86, 1.15)	10.66	−0.093	0.926	-	-
	DTaP–HBV–IPV–Hib vs. DTaP–IPV–Hib+HBV	1	1.11 (0.94, 1.3)	7.07	1.256	0.209	-	-
	Overall	12	1.03 (0.98, 1.07)	100.00	1.188	0.235	0.00%	0.671
Restlessness								
	DTaP–IPV–Hib vs. DTaP–IPV+Hib	3	0.96 (0.84, 1.1)	16.24	−0.579	0.562	68.40%	0.042
	DTaP–HBV–Hib vs. DTaP–HBV+Hib	3	0.96 (0.91, 1.01)	64.16	−1.475	0.14	8.90%	0.334
	DTaP–HBV–IPV–Hib vs. DTaP–HBV–IPV+Hib	1	0.96 (0.85, 1.08)	19.59	−0.675	0.500	-	-
	Overall	7	0.96 (0.92, 1.01)	100.00	−1.687	0.092	29.60%	0.202
Sleepiness								
	DTaP–IPV–Hib vs. DTaP–IPV+Hib	2	0.97 (0.87, 1.08)	29.33	−0.528	0.597	61.60%	0.106
	DTaP–HBV–Hib vs. DTaP–HBV+Hib	1	0.99 (0.94, 1.04)	64.69	−0.402	0.688	-	-
	DTaP–HBV–IPV–Hib vs. DTaP–HBV–IPV+Hib	1	1.02 (0.8, 1.3)	5.98	0.127	0.899	-	-
	Overall	4	0.99 (0.94, 1.04)	100.00	−0.585	0.559	0.00%	0.438
Unusual drowsiness								
	DTaP–IPV–Hib vs. DTaP–IPV+Hib	5	0.95 (0.85, 1.06)	31.28	−0.912	0.362	36.90%	0.175
	DTaP–HBV–Hib vs. DTaP–HBV+Hib	3	0.94 (0.85, 1.04)	34.07	−1.16	0.246	57.60%	0.095
	DTaP–HBV–IPV–Hib vs. DTaP–HBV–IPV+Hib	1	1.02 (0.9, 1.16)	21.47	0.339	0.734	-	-
	DTaP–HBV–IPV–Hib vs. DTaP–IPV–Hib+HBV	1	0.97 (0.82, 1.14)	13.18	−0.412	0.681	-	-
	Overall	10	0.96 (0.91, 1.02)	100.00	−1.187	0.235	24.10%	0.222
Vomiting								
	DTaP–IPV–Hib vs. DTaP–IPV+Hib	4	1.05 (0.86, 1.28)	52.58	0.483	0.629	56.90%	0.073
	DTaP–HBV–Hib vs. DTaP–HBV+Hib	3	1.05 (0.91, 1.2)	47.42	0.653	0.514	30.90%	0.235
	Overall	7	1.07 (0.95, 1.21)	100.00	1.046	0.295	59.90%	0.020

## Data Availability

The data presented in this study are available in the article.

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
