# Peer review of "Immunogenicity and Safety of Childhood Combination Vaccines: A Systematic Review and Meta-Analysis"

_vaccines, 2022, doi:10.3390/vaccines10030472_

Round 1
Reviewer 1 Report
The article is interesting and is suitable for publication
Author Response
Thank you for your support.
Reviewer 2 Report
In the Systematic Review titled «Immunogenicity and safety of childhood combination vaccines: A systematic review and meta-analysis», the authors summarize the available information on childhood combination vaccines (DTaP-IPV-Hib, DTaP-HBV-Hib, and DTaP-HBV-IPV-Hib) and analyze the efficacy and safety of combination vaccines compared to separate vaccines. This systematic review includes all the phases required for this type of scientific research and may help in acquisition of new knowledge in this field of vaccination. However, the following concerns should be addressed prior to the publication.
- Please explain why there is a comparison between DTaP-HBV-Hib vs. DTaP-HBV+Hib vaccines that do not contain a polyo component (Table 2, column «Vaccine group», polio serotype 1, polio serotype 2, polio serotype 3)?
- The authors indicate that «the group that received DTaP-HBV-IPV-Hib had a higher risk of fever than the group that received DTaP-IPV-Hib + HBV (RR= 1.18; 95% CI: 1.04, 1.33; P =0.010) » (lanes 230, 231). However, as follows from Table 3, these data relate to the parameter «Irritability». The data need to be corrected. I also suggest that this issue be discussed in the text of this systematic review.
- The authors discuss that «The potential mechanism responsible for the immunogenicity differences among the combined and separate vaccines may be associated with the complex physical or chemical intermolecular interactions between the mixture component of combined vaccination». In this regard, I consider it appropriate to present the formulations of the compared vaccines with information on the adjuvants and excipients in Supplementary Materials.
- It would be interesting to know the authors' opinion on the relevance of comparing vaccine efficacy by focusing only on the humoral immune response, without evaluation the cell-mediated immunity.
- Please clarify the name of the combined vaccine DTaP-HBV-IPV/Hib (lanes 69, 70). Since this vaccine is described as a hexavalent vaccine, its full description should include the pertussis component.
- Please add the meaning of the abbreviation «aP» (acellular pertussis). I also think it is appropriate to note in the text that PT, FHA, and PRN are pertussis antigens.
- In Table 1 (studies 19-23, column "Vaccine comparisons"), replace "DTaP-HBV-IPV+HBV" with "DTaP-IPV-Hib+HBV". A similar replacement is needed in line 191. Why do combination vaccine names use the symbols either «-» or «/»? I recommend unifying the names.
Author Response
In the Systematic Review titled «Immunogenicity and safety of childhood combination vaccines: A systematic review and meta-analysis», the authors summarize the available information on childhood combination vaccines (DTaP-IPV-Hib, DTaP-HBV-Hib, and DTaP-HBV-IPV-Hib) and analyze the efficacy and safety of combination vaccines compared to separate vaccines. This systematic review includes all the phases required for this type of scientific research and may help in acquisition of new knowledge in this field of vaccination. However, the following concerns should be addressed prior to the publication.
Reply: We appreciate the reviewer’s positive evaluation of our work. Your precious comments and advice are really helpful for the improvement of this manuscript.
1.Please explain why there is a comparison between DTaP-HBV-Hib vs. DTaP-HBV+Hib vaccines that do not contain a polyo component (Table 2, column «Vaccine group», polio serotype 1, polio serotype 2, polio serotype 3)?
Reply: Actually DTaP-HBV-Hib vs. DTaP-HBV+Hib could provide the baseline data for comparison for the polio. Of course, we could delete it to avoid the confusion if the reviewer think it is necessary.
2.The authors indicate that «the group that received DTaP-HBV-IPV-Hib had a higher risk of fever than the group that received DTaP-IPV-Hib + HBV (RR= 1.18; 95% CI: 1.04, 1.33; P =0.010) » (lanes 230, 231). However, as follows from Table 3, these data relate to the parameter «Irritability». The data need to be corrected. I also suggest that this issue be discussed in the text of this systematic review.
Reply: Thank you for pointing this out. We have made the correction in Results section, as, “the group that received DTaP-HBV-IPV-Hib had a higher risk of irritability than the group that received DTaP-IPV-Hib + HBV (RR= 1.18; 95% CI: 1.04, 1.33; P =0.010).” (Page8, line251-253, marked in red) and added corresponding illustration in the Discussion section, as “Irritability was the most frequently reported solicited systemic reaction. The meta-analysis revealed that the group that received DTaP-HBV-IPV-Hib had a higher risk of irritability than the group that received DTaP-IPV-Hib + HBV (RR= 1.18; 95% CI: 1.04, 1.33; P =0.010).” (Page11, Line 331-334, marked in red )
3.The authors discuss that «The potential mechanism responsible for the immunogenicity differences among the combined and separate vaccines may be associated with the complex physical or chemical intermolecular interactions between the mixture component of combined vaccination». In this regard, I consider it appropriate to present the formulations of the compared vaccines with information on the adjuvants and excipients in Supplementary Materials.
Reply: We are grateful for the suggestion. The relevant information has been added in the Supplementary Material as follows:
Supple Table1 Antigenic content of some of the DTaP-based combined vaccines
|
Vaccine |
Trade name |
Manufacturer |
Antigen contents |
Age |
|||||
|
D |
T |
aP |
HBV |
Hib |
IPV |
||||
|
DTaP-HBV-IPV |
Pediarix |
GlaxoSmithKline |
√ |
√ |
√ |
√ |
- |
√ |
6 weeks to 6 years |
|
DTaP-IPV-Hib |
Pentacel |
Sanofi Pasteur |
√ |
√ |
√ |
- |
√ |
√ |
6 weeks to 4 years |
|
DTaP-HBV-Hib-IPV |
Infanrix hexa |
GlaxoSmithKline |
√ |
√ |
√ |
√ |
√ |
√ |
6 weeks to 2 years |
DTaP: Diphtheria, Tetanus, acellular Pertyussis; HBV: Hepatitis B Virus; Hib: Haemophilus influenzae type b; IPV: Inactivated Polio Virus.
Supple Table2 Antigen composition of DTaP-HBV-IPV-Hib
|
|
DTaP-HBV-Hib-IPV |
|
Diphtheria toxoid (DT) |
DT (>30 IU)absorbed on AI(OH)3 |
|
Tetanus toxoid (TT) |
TT (>40 IU)adsorbed on AI(OH)3 |
|
Pertussis antigens |
Pertussis toxoid (25μg), filamentous hemagglutinin(25μg), and pertactin(8μg), absorbed on AI(OH)3 |
|
Inactivated poliovirus (IPV) |
IPV type 1 (Mahoney strain, 40 D-antigen unit), type 2 (MEF-1 strain, 8D-antigen unit), type 3 (Saukett strain, 32 D-antigen unit) |
|
Hacmophilus influenzac typeb (Hib) |
Hib polysaccharide (polyribosylribitol phosphate, 10 µg) conjugated to tetanus toxoid (~25 µg), adsorbed on Al(PO4) |
|
Hepatitis B (HepB) |
HB surface antigen (10 µg) adsorbed on Al(PO4) |
Composition for 1 dose (0.5 mL for each vaccine). IU: international unit; Al(OH)3: aluminum hydroxide, hydrated; Al(PO4): aluminum phosphate
4.It would be interesting to know the authors' opinion on the relevance of comparing vaccine efficacy by focusing only on the humoral immune response, without evaluation the cell-mediated immunity.
Reply: We agree with you. The cell-mediated immunity is important for evaluating efficacy of vaccine. However, because of availability issue of cell-mediated immunity related data, most studies evaluation by using humoral immune response, therefore, we only examined the humoral immune response based on the published literatures.
5.Please clarify the name of the combined vaccine DTaP-HBV-IPV/Hib (lanes 69, 70). Since this vaccine is described as a hexavalent vaccine, its full description should include the pertussis component.
Reply: Thank you for pointing this out. We have made the correction in Introduction section, as, “The combined hexavalent diphtheria-tetanus-acellular pertussis-HepatitisB- inactivated poliomyelitis and H. influenzae type B conjugated vaccine (DTaP-HBV-IPV-Hib).......”. (Page2, Line 76-78, marked in red )
6.Please add the meaning of the abbreviation «aP» (acellular pertussis). I also think it is appropriate to note in the text that PT, FHA, and PRN are pertussis antigens.
Reply: Thank you for your suggestion. We have added relevant description in the Results section, as “The acellular pertussis(aP) antigens include purified pertussis toxin (PT), filamentous haemagglutinin (FHA) and pertactin (PRN). ” (Page7, Line 215-216, marked in red ).
7.In Table 1 (studies 19-23, column "Vaccine comparisons"), replace "DTaP-HBV-IPV+HBV" with "DTaP-IPV-Hib+HBV". A similar replacement is needed in line 191. Why do combination vaccine names use the symbols either «-» or «/»? I recommend unifying the names.
Reply: We apologize for the confusing description. We have unified the names in full text.
Reviewer 3 Report
Authors reviewed the immunogenicity and safety of childhood combination vaccines using a systematic review and meta-analysis.
Although this manuscript is potentially interesting, several issues arise.
Abstract. There were many abbreviations. Full name should be shown at first time.
Abstract. “-“, “+” and “/” should be explained.
Abstract. The efficacy should concretely be shown.
Many abbreviations in Tables and Figures should be explained in these legends.
Figure 2 is unclear.
Table 2. Immunogenicity should be explained.
Table 3 Were there serious adverse events?
Author Response
Authors reviewed the immunogenicity and safety of childhood combination vaccines using a systematic review and meta-analysis.
Although this manuscript is potentially interesting, several issues arise.
Reply: Thank you very much for your positive comments and constructive suggestions, which are really helpful for the improvement of this manuscript.
Abstract. There were many abbreviations. Full name should be shown at first time.
Reply: Thank you for your suggestion. We have added full name for abbreviation at the first time.
Abstract. “-“, “+” and “/” should be explained.
Reply: Thank you for pointing this out. We have unified the vaccine names in Abstract section. For clarity, we removed “+” and “/”.
Abstract. The efficacy should concretely be shown.
Reply: Thank you for your suggestion. We have added concrete description of the immunogenicity in Abstract section, as “The results indicated that the combined diptheria-tetanus-acellular pertussis(DTaP)-hepatitis B virus(HBV)-Haemophilus influenzae type B (Hib) vaccine group had lower levels of anti-tetanus antibodies than the combined DTaP-HBV and separate Hib vaccinations group (SMD = -0.23; 95% CI: -0.42, -0.05; P = 0.013). Meanwhile, the combined DTaP-HBV-inactivated polio virus(IPV)-Hib vaccine group had higher levels of anti-pertussis(PT) and anti-filamentous hemagglutinin(FHA) antibodies than the combined DTaP-IPV-Hib and separate HBV vaccinations group (anti-PT: SMD = 0.60; 95% CI: 0.45, 0.75; P < 0.0001; anti-FHA: SMD = 0.40; 95% CI: 0.01, 0.78; P = 0.042). The levels of anti-pertactin (PRN) antibodies were lower in the combined DTaP-IPV-Hib vaccine group than in the combined DTaP-IPV and separate Hib vaccinations group(SMD = -0.13; 95% CI: -0.27, -0.00; P =0.047). ” (Page 1, Line 30-39, marked in red ) .
Many abbreviations in Tables and Figures should be explained in these legends.
Reply: Thank you for pointing this out. We have added information for full name and description for abbreviation in footnote of Table2, as“Abbreviations: PT, pertussis; FHA, filamentous hemagglutinin; PRN, pertactin; PRP, polyribosyl ribitol phosphate; HBV, hepatitis B; DTaP, tetanus diphtheria, and acellular pertussis vaccine” (Page 8, Line 225-226, marked in red )
Figure 2 is unclear.
Reply: Thank you for pointing this out. To make it clearer, we have modified the Figure2. Hope it is clear now.
Table 2. Immunogenicity should be explained.
Reply: Thank you for pointing this out. We have explained the immunogenicity in footnote section of Table 2, as “immunogenicity: non-inferiority of genometric meantiters (GMTs); SMD /(GMTs) : standard mean difference of GMTs” (Page 8, Line 224, marked in red )
Table 3 Were there serious adverse events?
Reply: Thank you for pointing this out. We did not catch the serious adverse events in our study. We only analyzed common local and systemic adverse effects. Local adverse effects included pain, redness and swelling at the site of injection. Systemic adverse effects comprised of diarrhea, fever, irritability, loss of appetite, restlessness, sleepiness, unusual drowsiness and vomiting.
Round 2
Reviewer 3 Report
This manuscript has been sufficiently improved. I have no further comment.